# Statistical Analysis of Single-Order Diffraction Grating with Quasi-Random Structures

**Huaping Zang [1,*], Zhihao Cui [1,2], Lai Wei [2,*], Hongjie Liu [2], Quanping Fan [2], Yangfan He [2], Bin Sun [2,3], Jihui Chen [2,4] and Leifeng Cao [5]**

1 Key Laboratory of Material Physics, Ministry of Education, School of Physics and Microelectronics, Zhengzhou University, Zhengzhou 450052, China
2 Science and Technology on Plasma Physics Laboratory, Laser Fusion Research Center, China Academy of Engineering Physics, Mianyang 621900, China
3 Department of Plasma Physics and Fusion Engineering, CAS Key Laboratory of Geospace Environment, University of Science and Technology of China, Hefei 230026, China
4 Institute of Modern Physics, Fudan University, Shanghai 200433, China
5 School of engineering Physics, Shenzhen Technology University, Shenzhen 518118, China
* Correspondence: zanghuaping@zzu.edu.cn (H.Z.); weilai@caep.ac.cn (L.W.)

**Abstract:** Single-order diffraction gratings with quasi-random structures are effective optical elements in suppressing harmonics contamination. However, background intensity fluctuations introduced by quasi-random structures may affect the measurement of the spectra and the fluctuations lack quantitative description. A unified theoretical method is provided to describe quasi-random diffraction structures with arbitrary distribution functions and an arbitrary number of microstructures. The effect of the number of microstructures and distribution functions on the level of background fluctuations is evaluated. This work provides important guidance for the design and optimization of single-order diffraction gratings, which are attractive for spectral analysis and monochromator applications in synchrotron beam lines.

**Keywords:** high-order diffraction suppression; grating; spectroscopy





## 1. Introduction

Synchrotron beamlines play an important role in a wide range of fields [1–3], such as spectroscopic studies, polymer science in water windows band, biological and magnetic material research, quantitative high-precision wavelength metrology of various optical devices, and reflectometry techniques in a variety of scientific applications [1,4]. The signals generated by higher harmonics can mask or exceed the signal generated by the fundamental wave, affecting the purity of the spectrum. [5]. The reliability of experimental data will be significantly compromised in the absence of a high-harmonic suppression device on the beamline. The main methods used for the suppression of higher harmonics are filters, suppression mirrors, and gas absorption cells [1,6,7]. These methods may require additional suppression systems or optical elements for monochromatic applications [8]. Compared to other methods, single-order diffraction gratings can suppress higher harmonics without requiring additional equipment and cover a wide energy spectrum. Single-order diffraction gratings only have 0th and ±1st order diffractions, and their spectral positions strictly correspond to different wavelengths. [9]. In the far field of the visible light band, sinusoidal gratings are ideal single-order diffraction gratings that can provide a pure monochromatic light source [10]. However, in the X-ray band, sinusoidal gratings introduce phase shifts that may result in the re-appearance of higher order diffractions [5,11].

In recent research, Cao et al. [12] proposed a novel approach to address the issue of continuous phase shift in the X-ray range by introducing the concept of binarized transmittance [13,14]. The transmittance of a point on the grating is either 1 (completely transparent)

or 0 (completely opaque). Various designs of binarized gratings have been developed to achieve a binarized sinusoidal grating effect in the X-ray band [12]. These designs can be broadly classified into two types: periodic structures (such as tilted rectangular apertures, trapezoidal aperture gratings, etc.) [15,16], and quasi-random structures (such as spectroscopic photonic sieve, quantum dot grating, etc.)[17–20]. Quantum dot grating is a representative quasi-random design that uses a large number of microstructures in each period to replace the grating's grating bar structure. These microstructures achieve single-order diffraction by following a sinusoidal density distribution function.

The suppression of higher order diffractions by the periodic structure design depends on the particular shape and precise size of microstructures. Due to the proximity effect of the electron beam etching processes [21], the actual machined size is difficult to meet the requirements. The size deviation of the machined samples still surpasses 10% even with strict control of the process parameters at each step [22], which reintroduced higher order diffractions [15]. In contrast, the suppression effect of quasi-random structure designs on higher order diffractions is dependent on the distribution function [23], which is insensitive to the accuracy of single microstructure machining and has a higher error tolerance [24]. However, quasi-random structures lead to fluctuations in the background intensity [25], which may alter the spectral shape and significantly impact spectral analysis, particularly in broadband spectra measurement. Previous transmittance design and analysis methods have yet to systematically evaluate the effect of background intensity fluctuations on diffraction patterns [26].

In this paper, a statistical analysis method to describe the fluctuations of the average diffraction intensity and background intensity of quasi-random structures is proposed. The variation of the background intensity and the average diffraction intensity is examined in depth using the microstructure's number and distribution function.

## 2. Principles

The general diffractive grating can be regarded as a convolution of a periodic lattice and single microstructure. As shown in Figure 1a, $L_{pq}$ is a two-dimensional periodic lattice composed of $\delta$ functions, $(\mu_p, \nu_q)$ are the centers of the microstructures $(\mu_p, \nu_q) = (pd_1, qd_2)$, where $p$ represents the ordinal value of the rows of the lattices ($p = 0, \pm 1, \pm 2, \ldots, \pm P$), $q$ represents the ordinal value of the columns ($q = 0, \pm 1, \pm 2, \ldots, \pm Q$), $d_1$ and $d_2$ denote the periods of the array in the $\mu$ and $\nu$ directions, respectively. The total number of microstructures is $N = (2P + 1)(2Q + 1)$. The diffraction intensity of the grating is as follows:

$$
\begin{aligned}
I(x, y) &= |FT(g(\mu, \nu))|^2 * |FT(L_{pq})|^2 \\
&= I_0(x, y) \left| \sum_{(p,q)}^{N} \exp[-ik(\mu_p x + \nu_q y)] \right|^2,
\end{aligned}
\tag{1}
$$

where $x = \xi/z$, $y = \eta/z$, $(\xi, \eta)$ denotes the detection plane of diffraction pattern. $z$ denotes the distance between the diffraction grating and the diffraction pattern detection plane. * represents the convolution, $FT(\cdot)$ represents the Fourier transform, and $k$ is the wave number.$g(\mu, \nu)$ represents the transmittance of a single microstructure and $I_0(x, y)$ represents the far-field diffraction intensity of a single microstructure.

In the grating with the quasi-random structures, the $\delta$ function becomes some non-negative real distribution function $\Gamma(\mu_p, \nu_q)$. The distribution of the coordinates of the microstructures obeys a given function. A real element $(\mu_p, \nu_q)$ is an isolated value, but in theory, it can be considered as a continuous variable when a large number of microstructures obey the distribution function. The $\Gamma(\mu_p, \nu_q)$ introduces a modulation in the diffraction

pattern that can suppress the higher order diffractions [23]. The normalized probability density distribution function is as follows:

$$L_{pq} = \frac{1}{N \iint\limits_{(\mu_p,\nu_q)} \Gamma(\mu_p,\nu_q)d\mu_p d\nu_q} \Gamma(\mu_p,\nu_q) * \sum_{(p,q)} \delta(\mu_p - pd_1, \nu_q - qd_2).$$

(2)

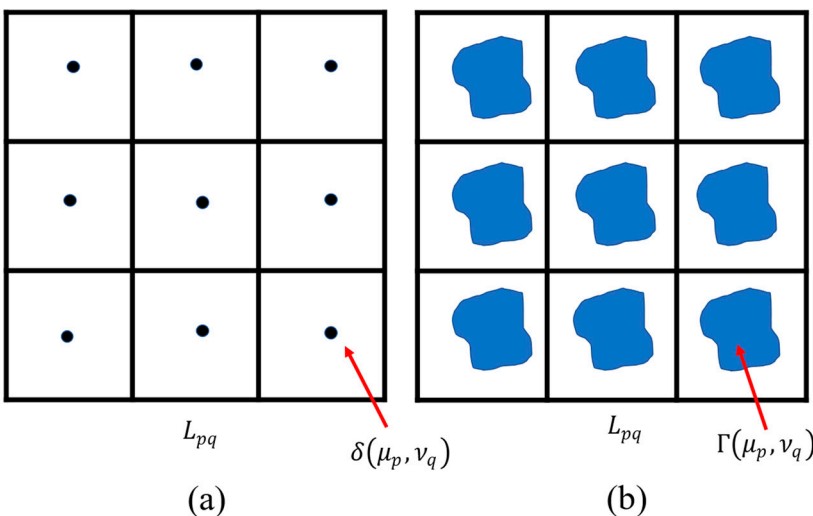

**Figure 1.** Schematic of the distribution of microstructures: from $\delta$ function to arbitrary positive function $\Gamma(\mu_p,\nu_q)$. (**a**) A two-dimensional periodic lattice composed of $\delta$ functions, (**b**) A two-dimensional periodic lattice composed of $\Gamma(\mu_p,\nu_q)$ functions.

If there is only one microstructure per period and $d_1 = d_2$ (single-microstructure structure, SMS), $|FT(L_{pq})|^2$ can be written as:

$$|FT(L_{pq})|^2 = N + \frac{1}{\left( \iint\limits_{(\mu_p,\nu_q)} \Gamma(\mu_p,\nu_q)d\mu_p d\nu_q \right)^2} \left| \int_{-d_1/2}^{d_1/2} \int_{-d_1/2}^{d_1/2} \Gamma(\mu_p,\nu_q) C_{pq} d\mu_p d\nu_q \right|^2$$
$$\sum_{(p,q)} C_{pq}(\mu_p,\nu_q) \sum_{(p\prime,q\prime)\neq(p,q)} C_{p\prime q\prime}^*(\mu_{p\prime},\nu_{q\prime}).$$

(3)

where the upper limit of all sums is N. The complex term $C_{pq} = \exp[ik(\mu_p x + \nu_q y)]$ and $C_{p\prime q\prime}^* = \exp[-ik(\mu_{p\prime} x + \nu_{q\prime} y)]$. Equation (3) contains two terms, the former term (N) represents the linear superposition of diffraction patterns of all microstructures, and the latter term includes interference effects between different microstructures.

When the grating has multiple microstructures in one period (multi-microstructures structures, MMS) that follow the same distribution function, the function $|FT(L_{pq})|^2$ of the grating needs to be reconsidered. The MMS can be decomposed into multi-layer SMS, and the distribution functions in different layers are identical, as shown in the Figure 2. Then, the $|FT(L_{pq})|^2$ of the grating with MMS is divided into two cases: the first case is the interference effect between different microstructures in the same layer; the second case is the interference effect between different microstructures in different layers. However, the case where two microstructures overlap must be excluded from consideration.

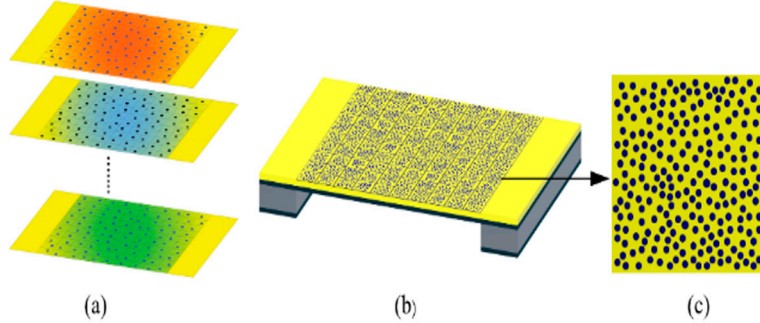

**Figure 2.** (**a**) Schematic diagram of the decomposition of the quasi-random MMS. (**b**) Schematic diagram of the quasi-random MMS. (**c**) Multi-microstructures in one period.

If the number of microstructures in a single period is M, the total number of microstructures is $N = M(2P+1)(2Q+1)$. The $\left|FT\left(L_{pq}\right)\right|^2$ of MMS can be written as follows:

$$
\left|FT\left(L_{pq}\right)\right|^2 = N + \frac{1}{\left(\underset{(\mu_p,\nu_q)}{\iint}\Gamma\left(\mu_p,\nu_q\right)d\mu_p d\nu_q\right)^2}\left|\int\limits_{-d_1/2}^{d_1/2}\int\limits_{-d_1/2}^{d_1/2}\Gamma\left(\mu_p,\nu_q\right)C_{pq}d\mu_p d\nu_q\right|^2 \times
$$

$$
\left(M\sum_{(p,q)}C_{pq}\sum_{(p',q')}C^*_{p'q'} + M(M-1)\sum_{(p,q)}C_{pq}\sum_{(p',q')\neq(p,q)}C^*_{p'q'}\right)
$$

$$
= N + B_1^2\left(M^2 D_1^2 - N\right)
$$

$$
B_1 = \frac{\left|\int\limits_{-d_1/2}^{d_1/2}\int\limits_{-d_1/2}^{d_1/2}\Gamma\left(\mu_p,\nu_q\right)C_{pq}d\mu_p d\nu_q\right|^2}{\left(\underset{(\mu_p,\nu_q)}{\iint}\Gamma\left(\mu_p,\nu_q\right)d\mu_p d\nu_q\right)^2},
$$

$$
D_1 = N\frac{\text{sinc}((2P+1)h)}{\text{sinc}(h)}\frac{\text{sinc}((2Q+1)l)}{\text{sinc}(l)},
$$

$$(4)$$

where *h* represents the transverse diffraction orders $h = xd_1/\lambda$, *l* represents the longitudinal diffraction orders $l = yd_1/\lambda$, $\lambda$ is the incident wavelength.

Therefore, according to Fraunhofer diffraction theory, the average diffraction intensity of any quasi-random MMS can be generalized to:

$$
\overline{I(x,y)} = I_0(x,y)\left(N + B_1^2\left(M^2 D_1^2 - N\right)\right),
$$

$$(5)$$

The same distribution function can produce different grating samples with quasi-random structures, and the diffraction intensity is different for different samples. In statistical analysis, the standard deviation is often used to measure the degree of fluctuation between a set of random variables and their average. The standard deviation distribution reflects the deviation of the diffracted intensity from its average intensity in different samples at a certain location, while the standard deviation distribution also reflects the fluctuation of the background intensity of a given sample. The standard deviation of the distribution can be written as:

$$
\sigma(I(x,y)) = \sqrt{\overline{I^2(x,y)} - \overline{I(x,y)}^2}.
$$

$$(6)$$

## 3. Results and Discussion

### 3.1. Quasi-Random Rectangular Distribution

As shown in Figure 3a, a rectangular region with equal side lengths $b = d/2$ is introduced at the center of a single period, and the microstructure is randomly distributed

within it. Here, the microstructure is a square hole. The distribution function can be expressed as:

$$\Gamma(\mu_n, \nu_n) = \text{rect}(\mu_n/b)\text{rect}(\nu_n/b). \tag{7}$$

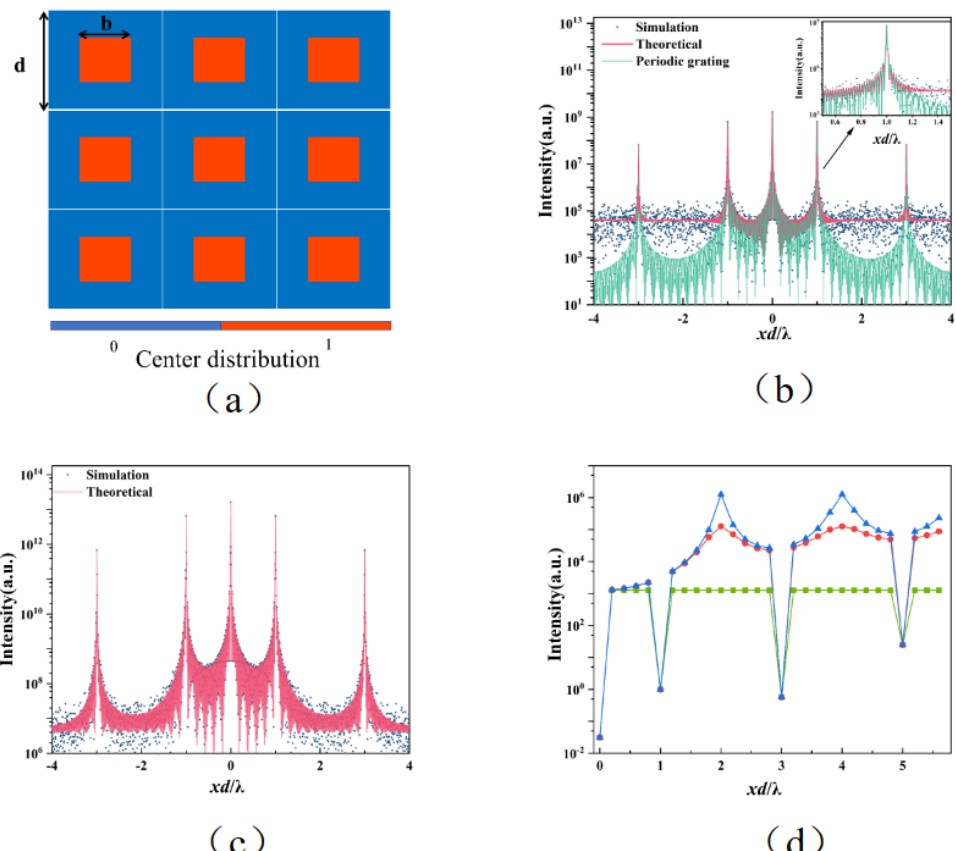

**Figure 3.** (**a**) Schematic diagram of Equation (7) (**b**) Theoretical and simulation of a SMS with rectangular distribution and a periodic grating with 1:1 duty cycle. (**c**) Theoretical and simulation of MMS (M = 100) with rectangular distribution. (**d**) Intensity ratio of quasi-random MMS with rectangular distribution and different number of square holes, the period number is 201 × 201 and the side length of the square hole is 0.01 d.

The far field average diffraction intensity of the quasi-random rectangular distribution structure can be obtained by combining Equations (3) and (5), where $B_1$ and $I_0$ are as follows:

$$B_1 = \text{sinc}(0.5h)\text{sinc}(0.5l) \tag{8}$$

$$I_0(h, l) = \sin c(sh)\sin c(sl), \tag{9}$$

In addition, $s$ is the side length of the square hole. The theoretical (Equation (3)) and simulation (Equation (1)) results of the average diffraction intensity of the quasi-random SMS and MMS (M = 100) with rectangular distribution are given in Figure 3b,c, respectively. The results show that the theory agrees well with the simulation results.

It is well known that a quasi—random structure grating can effectively suppress high order diffractions [25]. As shown in Figure 3b, by comparing the diffraction pattern of the SMS and a periodic grating with a duty cycle ratio of 1:1, we find that the high order diffraction intensity level of the two gratings is very similar with only odd-orders diffractions. However, the average background intensity of the quasi-random structure is higher than that of the periodic structure because some of the energy of the incident light will be transferred to the background from the high order diffraction peak. We defined "intensity ratio" as the ratio of first order average diffraction intensity to higher orders (>1st)

and background average intensity (non-diffraction peaks) to evaluate the suppression effect of the number of square holes within a period for intensities at different locations.

$$\frac{\overline{I(1,0)}}{\overline{I(h,0)}} \approx \begin{cases} \left[\frac{1+(N-1)\sin c^2(0.5)}{1+(N-1)\sin c^2(0.5h)}\right]\frac{I_0(1,0)}{I_0(h,0)}, & \{h|h=2n+1, n \in Z\} \\ \left[\frac{1+(N-1)\sin c^2(0.5)}{1-\sin c^2(0.5h)}\right]\frac{I_0(1,0)}{I_0(h,0)}. & \{h|h \neq 2n+1, n \neq 0\} \end{cases} \tag{10}$$

The ratio of the intensity of the 1st order to the 3rd and 5th order of the periodic grating is 9 and 25, respectively [25]. For quasi-random structures, the $\frac{I_0(1,0)}{I_0(h,0)}$ is approximated as 1 when the square hole size is much smaller than the period length. According to Equation (10), for a SMS, the intensity ratio of 1st to 3rd order intensity is 9, and the intensity ratio of 1st order to the background ($h = 2$, $l = 0$) intensity is $1.6 \times 10^4$. For MMS (M = 100), the intensity ratios of the 1st order to the 3rd order intensity and the background ($h = 2$, $l = 0$) intensity are 9 and $1.6 \times 10^6$, respectively. It is estimated from Equation (11) that when N >> 1, the quasi-random structure of the rectangular distribution has an intensity ratio $\sim h^2$ for each odd-order (>1st) and ~N for the background. As shown in Figure 3d, when increasing the number of square holes in a single period, the suppression of odd higher order (>1st) diffractions has no significant change, while the suppression of the background is obviously optimized, which is in accordance with the above theory.

However, the number of square holes cannot be indefinitely increased. The currently used research CCD has a dynamic range of approximately $2^{12} \sim 2^{16}$, and the intensity range of the diffraction pattern of MMS (M = 100) is around $2^{19}$ (average diffraction intensity of the first order to the average background intensity). Consequently, the average intensity of the background introduced by the quasi-random structure is lower than the noise level of a CCD. Therefore, the background introduced by the quasi-random structure can be ignored when M > 100.

The background fluctuations of the quasi-random structure are unavoidable. From Equation (6), the expression for the standard deviation distribution is

$$\sigma(I(x,y)) == \sqrt{I_0^2(x,y)[N^2 A_s + N B_s + C_s]}, \tag{11}$$

where $D_1$ is from Equation (5) and $B_1$ is from Equation (8), and $A_s, B_s, C_s$ are as follows:

$$\begin{cases} A_s = (1 - 2B_1^2 + B_1^4) \\ B_s = B_1^4(-6 - 2D_1^2) + B_1^2(2D_1^2 + 4B_2 + 4) - B_2^2 - 1 \\ C_s = B_1^4[D_1^2(7 - 2D_2) + 2D_2^2] + B_1^2[D_1^2(-4 - 4B_2 + 2B_2 D_2) - 2B_2 D_2^2] + B_2^2 D_2^2 \\ B_2 = \text{sinc}(h)\text{sinc}(l) \\ D_2 = \frac{\sin((2P+1)2h\pi)}{\sin(2h\pi)} \times \frac{\sin((2Q+1)2l\pi)}{\sin(2l\pi)} \end{cases}, \tag{12}$$

In Figure 4a,b, the theoretical average diffraction intensity and standard deviation distribution are compared for SMS and MMS (M = 100). The results indicate that the average diffraction intensity and standard deviation of MMS are higher than SMS. To assess the intensity fluctuations in the diffraction pattern, when the number (M) of square holes in a single period increase. The relative fluctuations were defined, which is the ratio of standard deviation to the average diffraction intensity at the same position and can be written as:

$$\frac{\sigma(h,0)}{\overline{I(h,0)}} \approx \begin{cases} \frac{\sqrt{N^3}}{N+N(N-1)\sin c^2(0.5h)}, & \{h|h=2n+1, n \in Z\} \\ \frac{1}{1-\sin c^2(0.5h)}, & \{h|h \neq 2n+1, n \neq 0\}. \end{cases} \tag{13}$$

Figure 4c demonstrates that the relative fluctuations at the odd-order positions of the MMS rectangular distribution are smaller compared to the background positions, and these fluctuations decrease with an increase in the number of square holes. This indicates that the intensity differences of the odd-orders between different MMS samples are

smaller. Figure 4d shows that the odd-order (>1) diffraction suppression effect of the MMS rectangular distribution is independent of the number of holes. The relative fluctuations of the quasi-random MMS with rectangular distribution are larger at the background position, and these fluctuations largely remain the same as the number of square holes increases (non-diffraction peak). Moreover, the intensity fluctuation of the MMS with rectangular distribution is more apparent in the background than in the diffraction peaks, as shown in Figure 4c. Accounting for the intensity fluctuation of the background in the actual diffraction pattern, the diffraction intensity range of MMS is $2^{16} \sim 2^{23}$. The background intensity is still lower than the noise level of the CCD and can be ignored.

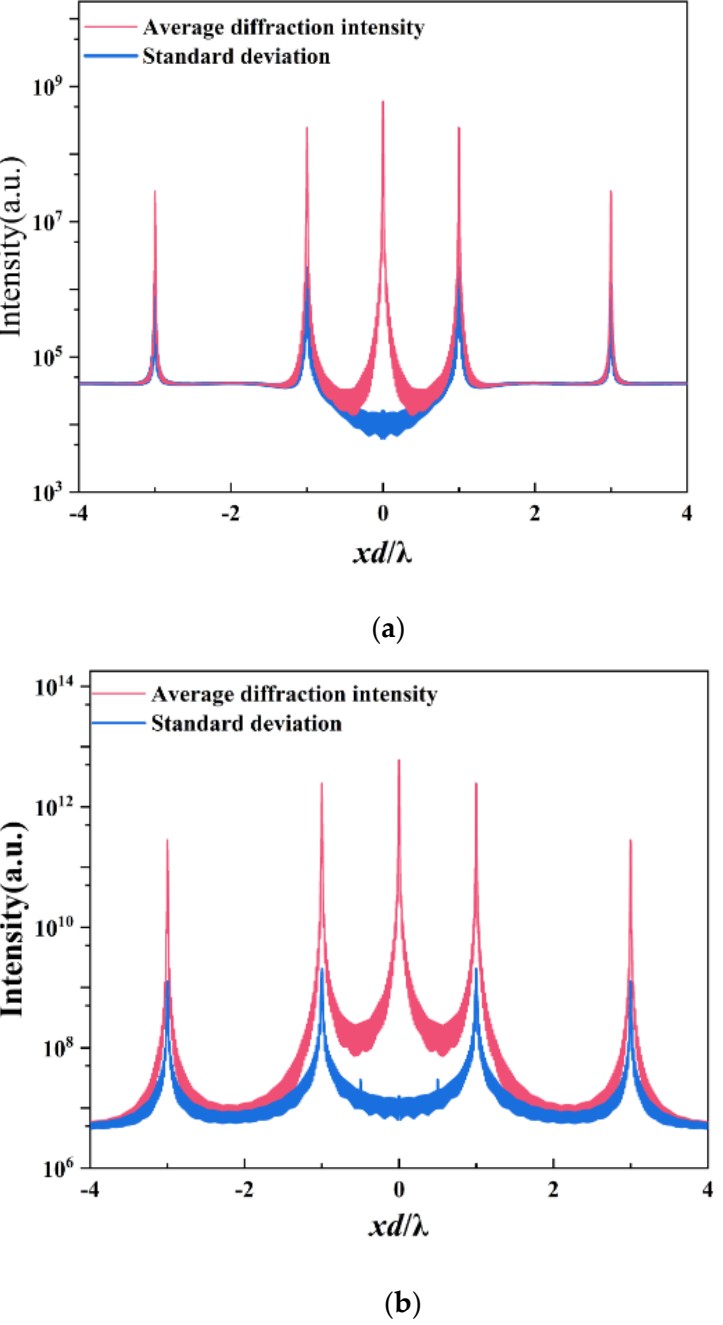

**Figure 4.** *Cont.*

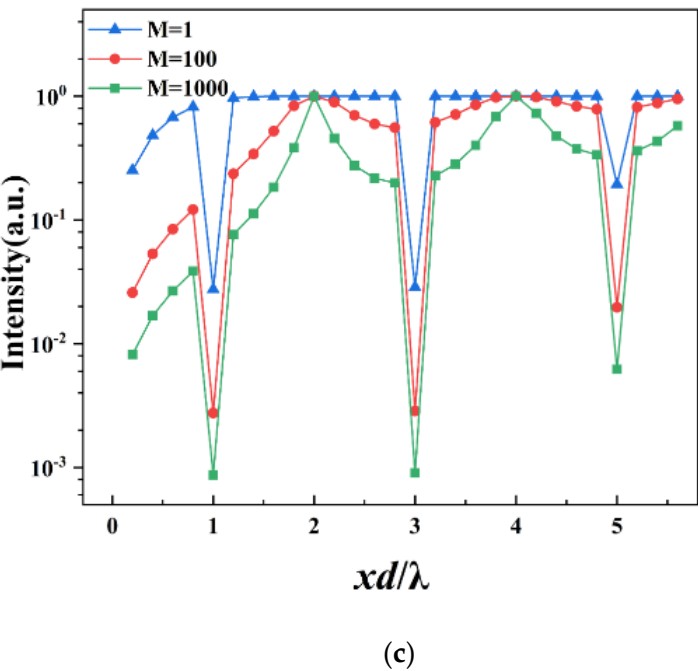

**(c)**

**Figure 4.** (**a**) Theoretical average intensity and standard deviation distribution of the quasi-random SMS with rectangular distribution. (**b**) Theoretical average intensity and standard deviation distribution of the quasi-random MMS (M = 100) with rectangular distribution. (**c**) Relative fluctuations of the quasi-random MMS with rectangular distribution of a different number of square holes, when the period number is 201 × 201 and the side length of the square hole is 0.01 d.

### 3.2. Quasi-Random Sinusoidal Distribution

As shown in Figure 5a, the distribution probability of the locations of the centers of the microstructure within the period follows the sinusoidal distribution function, and the microstructure is also a square hole. The distribution function can be expressed as:

$$F(\mu_n, \nu_n) = [\frac{1}{2} + \frac{1}{2}\cos(\frac{2\pi\mu_n}{d})]. \tag{14}$$

The average diffraction intensity and standard deviation distribution of the quasi-random sinusoidal structure can also be described by Equations (3) and (11), except that $B_1$ and $B_2$ are:

$$\begin{cases} B_1 = \text{sinc}(l)[\text{sinc}(h) + 0.5\text{sinc}(h+1) + 0.5\text{sinc}(h-1)] \\ B_2 = \text{sinc}(2l)[\text{sinc}(2h) + 0.5\text{sinc}(2h+1) + 0.5\text{sinc}(2h-1)]. \end{cases} \tag{15}$$

The theoretical and simulation results of the sinusoidal distribution structure are also consistent, as shown in Figure 5b,c, and it can be seen that the quasi-random structure grating with sinusoidal distribution only has diffraction of 0 and 1st order. The following equation illustrates the intensity ratio between the higher orders and the background average diffraction intensity of the quasi-random structure with a sinusoidal distribution,

$$\frac{\overline{I(1,0)}}{\overline{I(h,0)}} \approx \begin{cases} 1, & \{h|h=1\} \\ \left[\frac{1+0.25(N-1)}{1-[\text{sinc}(h)+0.5\text{sinc}(h+1)+0.5\text{sinc}(h-1)]}\right]\frac{I_0(1,0)}{I_0(h,0)}, & \{h|h\neq 1\}. \end{cases} \tag{16}$$

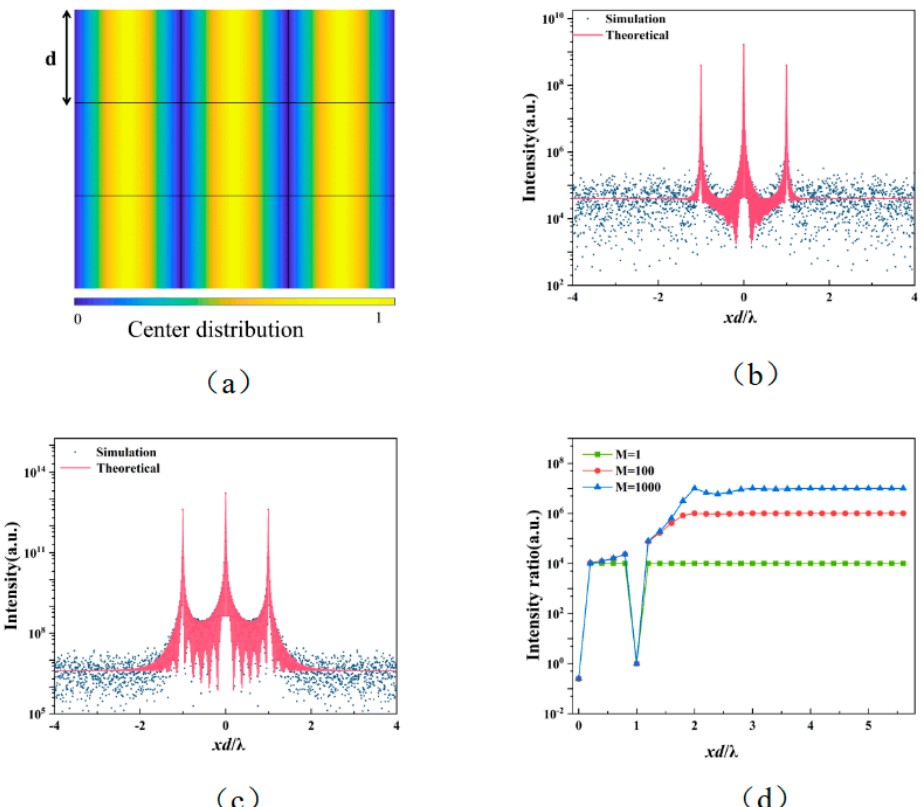

**Figure 5.** (**a**) Schematic diagram of Equation (14) (**b**) Theoretical and simulation of a SMS with sinusoidal distribution. (**c**) Theoretical and simulation of MMS (M = 100) with sinusoidal distribution. (**d**) Intensity ratio of quasi-random MMS with sinusoidal distribution different number of square holes, the period number is 201 × 201 and the side length of the square hole is 0.01 d.

The relative fluctuations of the quasi-random structure with sinusoidal distribution can be written as:

$$
\frac{\sigma(h,0)}{\overline{I(h,0)}} \approx 
\begin{cases}
\frac{\sqrt{N^3}}{N+0.25N(N-1)}, \{h|h=1\} \\
1 - [\mathrm{sinc}(h) + 0.5\mathrm{sinc}(h+1) + 0.5\mathrm{sinc}(h-1)], \{h|h \neq 1\}.
\end{cases}
\tag{17}
$$

In Figure 6a,b, the theoretical average diffraction intensity and standard deviation distribution are also compared for SMS and MMS (M = 100) with sinusoidal distribution. Figure 5d shows that the intensity ratio of the higher orders to the background follows ~ N. The relative fluctuations of the ±1 orders decrease with an increase in the number of square holes and are small (close to 0), whereas the relative fluctuations in the background remain relatively large (close to 1) and constant with an increase in the number of holes, as seen in Figure 6c. The fluctuations in the background are more obvious, consistent with the preceding analysis shown in Figure 5b,c. Taking the fluctuations into account, the diffraction intensity range of MMS (M = 100) with sinusoidal distribution is $2^{13} \sim 2^{23}$. Despite the existence of fluctuations, the background intensity is still lower than the noise level of the CCD, and it can be ignored in most situations.

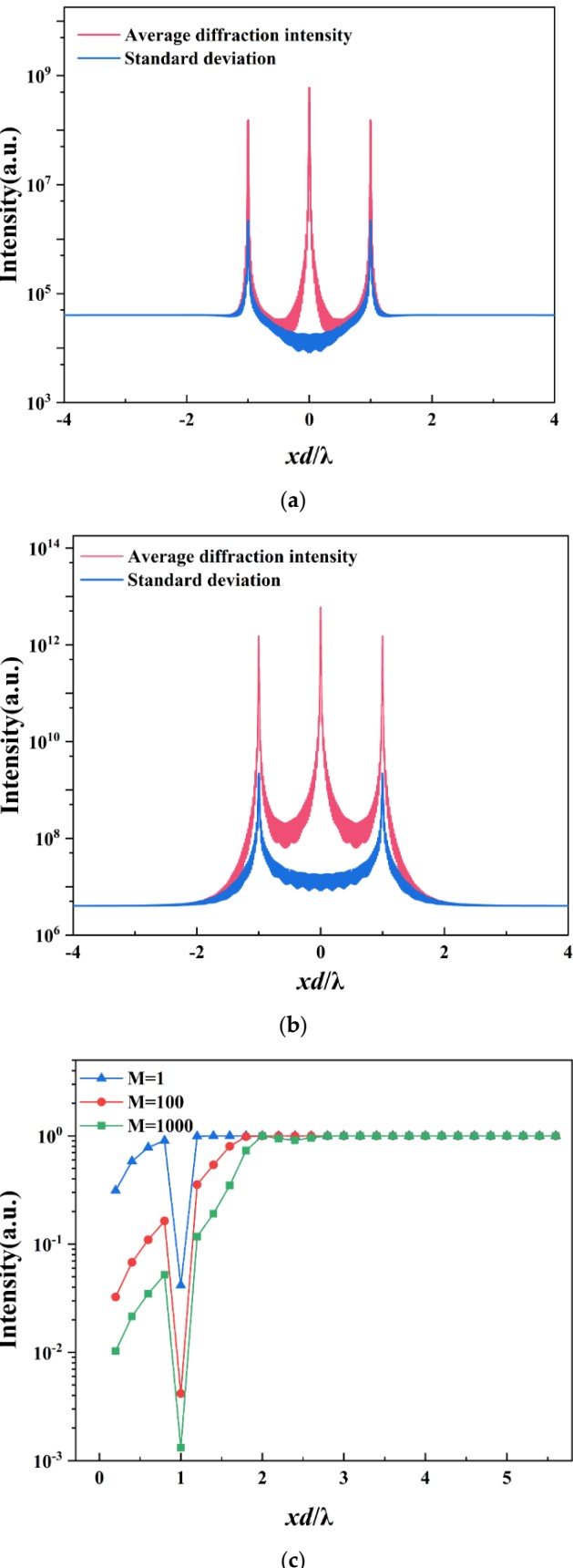

**Figure 6.** (**a**) Theoretical average intensity and standard deviation distribution of the quasi-random SMS with sinusoidal distribution. (**b**) Theoretical average intensity and standard deviation distribution

of the quasi-random MMS (M = 100) with sinusoidal distribution. (**c**) Relative fluctuations of the quasi-random MMS with sinusoidal distribution different number of square holes, when the period number is 201 × 201 and the side length of the square hole is 0.01 d.

## 4. Conclusions

In conclusion, a quasi-random MMS theory is proposed based on the quasi-random SMS theory, which describes the average diffraction intensity and fluctuations of the quasi-random structure. Theoretical and simulation analyses demonstrate that increased single-period microstructures can enhance the suppression of background intensity and reduce intensity fluctuations induced by quasi-random structures. Additionally, the comparison of the quasi-random rectangular distribution function and the quasi-random sinusoidal distribution function confirms that the sinusoidal distribution function is more effective in suppressing higher-order diffractions. Meanwhile, it is found that the background fluctuations introduced by the quasi-random structure can be ignored when M > 100 in the actual diffraction patterns. Quasi-random MMS theory can optimize the structural design of single-order diffraction gratings and has the potential to be extended to arbitrary quasi-random structure diffraction elements. This can facilitate the advancement and application of single-order diffraction gratings in the fields of spectral diagnosis and soft X-ray monochromatization.

**Author Contributions:** Simulation, investigation, writing, H.Z.; methodology, Q.F., Z.C. and J.C.; writing—review and editing, L.W.; visualization, Y.H. and B.S.; supervision, L.C. and H.L.; Funding acquisition, H.Z. L.W. and Q.F. All authors have read and agreed to the published version of the manuscript.

**Funding:** This research was funded by the National Natural Science Foundation of China (Grant Nos. 12174350, 12275250 and 12275253), Science and Technology on Plasma Physics Laboratory (Grant No. 6142A04200107).

**Institutional Review Board Statement:** Not applicable.

**Informed Consent Statement:** Not applicable.

**Data Availability Statement:** Data underlying the results presented in this paper are not publicly available at this time, but may be obtained from the authors upon reasonable request.

**Conflicts of Interest:** The authors declare no conflict of interest.

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
