# Peer review of "Statistical Analysis of Single-Order Diffraction Grating with Quasi-Random Structures"

_photonics, doi:10.3390/photonics10030303_

Round 1

Reviewer 1 Report

For this paper, the interesting working about the theory of MMS was present. there are two sugestion:

1. In an scientifical paper, the first person should be avoided.

2. In the introduction, there should be introduction about MMS, especially other alike researching .

Reviewer 2 Report

In this paper, the authors introduced a theoretical framework for analyzing quasi-random grating structure diffraction properties. The study was rigorously and systematically conducted and the conclusion is clear. Overall, I would recommend the manuscript to be published in Photonics if several following ambiguities can be addressed.

 1. In equation 1, the authors should explain the operator * (multiply) and I(far-field diffraction of the microstructure) for readers.

2. The sentence "The microstructures are randomly distributed in their respective periods and obey the same distribution function ..." is confusing - the microstructure are randomly distributed at different positions? Please also explain Figure 1 (b), in which the distribution function in each unit cell is not centered. 

3. Can the authors explain Figure 4(b) more? Please provide zoomed-in figures around the diffraction peaks for more clear presentations. In addition, the benefits of using quasi-random grating over periodic grating is not straightforward to me, please explain more.

4. Can the authors justify/explain using h=0.3 l=0 as the background intensity when calculating the suppression ratio?

Author Response

Please see attachment。

Reviewer 3 Report

  In the manuscript by Huaping Zang et al., a theory describing the diffracted intensity of a grating with quasi-random structures is described and applied to some simple model systems.

 The manuscript is not in a state suitable for review, and therefore I cannot judge whether the contents are correct or warrant publication. Especially, many symbols and mathematical notation are not defined, which makes it impossible to follow the equations without guessing what the authors could mean. In addition, some of the equations in the Principles section seem to be incorrect. The authors might have intended to write a correct equation, but the actual notation does not seem to make sense. I give examples of the problems in the first part of the manuscript below.

 The manuscript must be revised to conform with scientific standards and mathematical rigor before resubmission. Please note again that I was not able to conduct a full review and the comments below are just intended to show some of the problems. It is not the job of the reviewer to point out all problems of a manuscript that is in a state like this, the authors must thoroughly check the manuscript themselves.

Regarding equation (1) and following explanation:

-       - Define the * symbol. Could be convolution or simple multiplication.

-       - Define k

-       - Shouldn’t Ln , μn , νn etc also have the index m? Likewise, the sum in (1) goes over n and m?

-       L. 73: what is the meaning of  “denotes the receiving screen”? Coordinates?

-       L. 77: The sum over n in (1) is up to N, but N itself is dependent on n. This does not make sense.

-       L. 77: Explain why even numbers of rows and columns are excluded.

-       L. 78: ±n (±m) is given as the maximum value for n (m). This does not make sense.

-       - It would be helpful to explain that I0 is the diffracted intensity of one microstructure (if that is correct).

Regarding equation (2) and following explanation:

-       - In (2) the sum is over n and m, but μn , νn only depend on n?

-       - n is the subscript of Ln , but also used in the sum. Is this the same?

-      -  An explanation that μn , νn changed from single values to continuous variables would be helpful.

Regarding equation (3) and following explanation:

-       Define Cm* (complex conjugate?)

-       E is described as “the normalized joint probability density distribution function". Isn’t it the Fourier transform?

-       Writing E(Cn Cm*) means that E is a function of Cn  and Cm*, but they are not variables. This is confusing. In addition, Cm* appears only in the sum, so E really can’t be said to depend on Cm*

-       E and therefore I(x,  y) seem to depend on n and m, but which values should be used for them? I think I(x,  y) should not depend on n and m.

Regarding equation (4)

-       The second sum is over n’,  m’, but they don’t appear anywhere in the sum.

-       It is unclear whether the two sums are intended to multiplicated, or whether the second sum is inside the first sum (because only Cm* depends on m).

Equation (5) has similar problems.

-       In addition, why are the indices n and m in D1?

-       For the “diffraction order”, integer values would be expected. But in the manuscript this seems to be a continuous variable

Regarding Fig. 3 and the explanation of variance:

-       I can’t understand Fig. 3.

- Are the intensities for the different samples shifted? If yes, this should be mentioned.

- How do the rectangles correspond to the variance? It can’t be the height, because the variance has a different unit of the intensity. Even assuming that the standard deviation is really meant, it should not be so large (if the intensities for the different samples are shifted). It can’t be the area either, that does not make sense.

- Intensity values at different “diffraction orders” are compared, this does not correspond to equation (6).

- Why is the width of the rectangles different?

- Why are these four rectangles shown? Are other points not taken into account for the “distribution of variance”.

-       Is the variance used here different from the standard meaning? If yes, this should be explained. If no, is Fig. 3 really needed (as I mentioned above, I don’t understand it)?

Other comments:

-       I don’t think that the term “suppression ratio” is appropriate for the quantity defined in the manuscript, this is just the ratio of two intensities. For the “suppression ratio”, one would expect a suppression of the intensity with regard to a periodic grating, for example.

-       In Fig. 4 (d), the “suppression ratio” of a periodic grating should also be shown.

-       The “suppression ratio” in Fig. 4(d) is close to 1 for the 3rd harmonic. So there is no significant suppression of higher harmonics compared to a periodic grating?

-       It does not make sense to show the variance in Fig. 5 etc., because it does not have the same units as the intensity.

-       How is the orientation of the grating and the screen with regard to the incident beam taken into account? The grating will not be perpendicular to the incident beam in many experimental setups.

-       The statements regarding the use of gratings (l. 28) and the higher harmonics (l. 32) are known for a long time. Why are the references 2,3 and 6, 7, 8 respectively used, which are quite recent and specific papers? A textbook, review paper or the works first mentioning these would be appropriate.

-       L. 29: monochromatic synchrotron radiation beams are more often used to probe a sample than “to calibrate various scientific instruments”.

-       In what sense is the theoretical model “unified”? This is claimed in the abstract, but not explained or mentioned in the text.

-       There are numerous mistakes in the English. Proofreading by someone with a good command of English is needed.

Reviewer 4 Report

The manuscript "Statistical analysis of single-order diffraction grating with quasi-random Structures" by Huaping Zang et al. reports on a mathematical consideration of a 2D quasi-periodic grating monochromator for more efficient higher harmonics suppression in modern synchrotron beamline instrumentation.

The topic is generally suitable for Photonics. The manuscript is undoubtedly original and of potential interest to the broad readership. In my opinion, in the present form, it is too abstract. It is recommended to strengthen the connection with current instrumentation development challenges in the Introduction section of the manuscript. Please, specifically address the following points.

1) VUV-soft X-ray photon energy range is envisaged

2) The title optical element is rather a hybrid of a diffraction grating monochromator and a zone plate

3) It is the energy resolution that is the most essential characteristics of a monochromator. Please, add a concise discussion on the energy widths of allowed transmitted harmonics as a function of the number of microstructures.

4) The title optical element is expected to act as a focusing lense in addition to monocromatization. Please, address focusing properties of this quasi-periodic 2D grating monochromator as well.

5) Have there been any attempts to manufacture such a grating device and verify theoretical predictions on its properties experimentally?

Round 2

Reviewer 3 Report

The manuscript has improved, but there are still some problems. Please see the attached file for details. In particular, there are inconsistencies between Fig. 4 and the text, and between the subfigures in Fig. 4 and Fig. 5. Therefore some results seem to be wrong.

In addition, there appears to be an issue with authorship. Why is H. Z. the leading author? They have only contributed by funding acquisition according to the “Author Contributions”. This does not qualify to be an author according to the guideline of MDPI or international standards.

https://www.mdpi.com/journal/photonics/instructions#authorship
The leading author would usually be understood to have contributed most to the research and writing.

Author Response

Please see attachment。

Round 3

Reviewer 3 Report

I think that the mayor issues with the manuscript have been resolved. Please consider the two points below, however.

In addition, there are still errors  in English grammar and style.

L. 103. “The complex term ... are not independent variables”:

In the first version of the manuscript, Eq. 3 was written in a way that suggested that Cpq were independent variables. This problem has been removed, so I think that this sentence is not necessary.

L. 157: A sentence starting with “It is well known...” asks for a citation.

Author Response

Dear reviewer,

 Thank you for taking the time to review my article and provide feedback. I have made the changes to the article based on your suggestions. We checked the article for English and polished it.  I hope that you can review it again, if you have any further questions or suggestions, you can contact me.

(1) Thank you for pointing out this problem in our manuscript, we have removed this sentence.

(2) Following your suggestion, we have added a reference.

Thank you again for your valuable input and I wish you all the best.